# Nonlinear Dynamics and Performance Analysis of a Buck Converter with Hysteresis Control

Carlos I. Hoyos Velasco [1], Fredy Edimer Hoyos Velasco [2,*] and John E. Candelo-Becerra [2]

[1] Istituto Motori IM, Italian National Research Council, Via Guglielmo Marconi, 4, 80125 Napoli, Italy; c.i.hoyosvelasco@gmail.com

[2] Department of Electrical Energy and Automation, Faculty of Mines, Universidad Nacional de Colombia, Sede Medellín, Carrera 80 No. 65-223, Robledo, Medellín 050041, Colombia; jecandelob@unal.edu.co

\* Correspondence: fehoyosve@unal.edu.co

**Abstract:** This paper presents the mathematical modeling and experimental implementation of a Buck converter with hysteresis control. The system is described using a state-space model. Theoretical and simulation studies show that the zero hysteresis control leads to an equilibrium point with the implication of an infinite commutation frequency, while the use of a constant hysteresis band induces a limit cycle with a finite switching frequency. There exists a tradeoff between voltage output ripple and transistor switching frequency. An experimental prototype for the Buck power converter is built, and theoretical results are verified experimentally. In general terms, the Buck converter with the hysteresis control shows a robust control with respect to load variations, with undesired high switching frequency taking place for a very narrow hysteresis band, which is solved by tuning the hysteresis band properly.

**Keywords:** bifurcation; buck power converter; chaos; hysteresis control; nonlinear dynamics; switching frequency



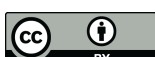

## 1. Introduction

The Buck converter is a system of variable structure [1] widely used in different technological applications [2], such as power sources for computer processors [3], battery charger modules [4], power supply of photovoltaic and inverter arrays [5,6], and a wide range of applications in automotive technology [7,8] to drive low voltage loads. However, this power converter presents nonlinear effects that must be studied to improve the system performance [9–11].

The application of hysteresis to the control loop increases complexity and provides similarity to other systems. For example, in mechanical systems, the hysteresis behavior can induce oscillations, impacting its precision and stability [12], while hysteresis can be applied to water temperature control [13]. In [14], the control with hysteresis [15,16] was proposed to reduce the commutation frequency of actuators, while in [17,18] has been recently applied by authors to address a regulation control problem in a boost power converter. In [19], authors state that the commutation frequency has an inverse relation to the amplitude of the hysteresis band.

Dynamic properties of a hysteresis model are presented in [20] for a piece-wise linear circuit. Here, the model consists of the superposition of hysteresis operators, and authors show the effect of hysteresis parameter values in both control performance and frequency of the induced hysteresis cycle. In [21], the authors designed and implemented an electronic converter to analyze photovoltaic and inverters arrays. Moreover, authors in [22] showed that the insertion of a hysteresis control achieves transient states of low overshoot.

In [23], a new hysteresis control strategy is presented with output-capacitor-equivalent series-resistance ripple compensation. When step load change occurs, the closed loop feedback control adjusts the duty ratio. The results of the paper show that the proposed

control scheme gains faster transient response speed and better steady-state operation than traditional hysteresis without ripple compensation.

In [24], a photovoltaic (PV) stand-alone system using a dual inverter based double-band hysteresis adaptive sliding-mode control (ASMC) is presented. Moreover, an adaptive hysteresis band is used to adopt the desired dual inverter switching frequency. Furthermore, dual inverter switching technique is employed to generate multilevel inverters output voltage. The effectiveness of the control scheme is verified under steady/transient states and load disturbance.

On the other hand, digital signal processing (DSP) provides greater flexibility than analog designs [25]. Some advantages are that the straightforward implementation of nonlinear controllers, quasi-sliding mode control (QSMC), and advanced control techniques [26]. However, quantization and discretization can cause periodic bands [27], chaotic behavior, oscillations, subharmonics, and limit cycles [28]. Additionally, the time delays in the output of the controllers can produce instability [29]. Thus, the performance of a well-designed discretized quasi-sliding mode control (DQSMC) may be affected when executed by a digital controller [30] due to the finite sampling frequency.

In [31], the authors presented research related to bifurcation in second-order systems using pulse-width modulation (PWM) and zero average of the error dynamics [32]. Similarly, Burgos [33] presented the study of nonlinear phenomena present in the Buck converter when it is controlled by the side-pulse PWM; the control surface is made to comply with the requirement of zero in the error dynamics. Additionally, bifurcation diagrams have been drawn theoretically and experimentally to show the existence of chaos and its control through the fixed point induction control (FPIC) technique, allowing chaotic dynamics in the power converter to be avoided [34,35].

All these works justify the development of a more detailed study about different dynamic effects present in the Buck converter. The hysteresis control technique is widely used to control systems with slow dynamics such as temperature or liquid level. Hence, studying a simple control strategy, such as hysteresis control to drive the power transistor, allows a better understanding of induced nonlinear dynamics and effects of high switching frequencies driving such electronic power converters.

This study aims at characterizing nonlinear dynamics and the existence of limit cycles induced by the hysteresis control. Therefore, this paper describes the Buck power converter controlled by a constant hysteresis band using a piece-wise mathematical model. Some contributions of this work include the following:

1. The constant hysteresis control is applied to a nonlinear and fast dynamic power converter system.
2. A mathematical model is derived and used to characterize the nonlinear dynamics induced by the hysteresis control.
3. A cost-effective voltage control strategy is implemented using few analog electronic components.

## 2. Materials and Methods

### 2.1. Model of the Buck converter

Let us consider the circuit of a Buck power converter presented in Figure 1. This circuit includes a voltage source connected by a switch $S$ to a filter $LC$ and a load $R$ [36]. In detail, the voltage source provides the input voltage $V_{in}$ [V], $S$ is an ideal switch, $D$ is a diode, $L$ [H] is an inductor, $C$ [F] is a capacitor, and $R$ [Ω] is the power converter resistive load. Note that, depending on the application area, the power converter load $R$ could present different features and responses, from linear to nonlinear and resistive to inductive loads, e.g., electronic modules, direct current (DC) motors, etc. The diode $D$ provides a path for the inductor current $i_L$ when the switch $S$ changes its state from closed to open, while the diode remains in reverse bias when the switch $S$ is closed. This electrical circuit is also known as a step-down converter due to the output voltage $v_C$ being lower than the input voltage $V_{in}$. Considering that the Buck converter includes two components capable of

storing energy, for modeling purposes, the state variable is defined as $x = [v_C, i_L]$. Herein, the term $v_C$ [V] is the capacitor voltage and the term $i_L$ [A] is the inductor current.

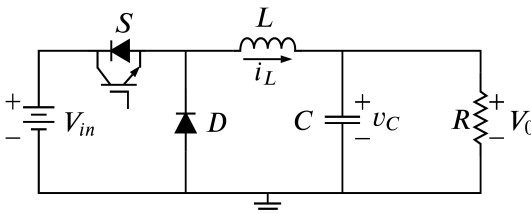

**Figure 1.** Electrical circuit of a Buck converter.

According to the status of the switch and diode, three topologies can be obtained, as shown in Figure 2 [37]. In particular, Figure 2a shows the first topology in continuous conduction mode (CCM) with positive inductor current $i_L > 0$, which occurs when the switch $S$ is closed, yielding a positive current through the inductor. Hence, the resultant circuit is reduced to the source with the input voltage $V_{in}$, an inductor $L$ in series with the source, a shunt capacitor $C$, and a shunt-load resistance $R$.

Additionally, Figure 2b shows the second topology evolving in CCM with $i_L > 0$, which occurs when the switch $S$ is open. Then, the inductor current continues to flow through the diode in forward bias until the inductor current becomes zero. The resultant circuit is reduced to three shunt elements composed of an inductor $L$, a capacitor $C$, and a resistance $R$. Finally, the third topology is shown in Figure 2c evolving in discontinuous conduction mode (DCM) with $i_L = 0$, which is formed when the switch $S$ is open and the inductor current is zero, due to the diode $D$ being deactivated or in reverse bias. The remaining RC circuit discharges the capacitor voltage through the resistive load $R$.

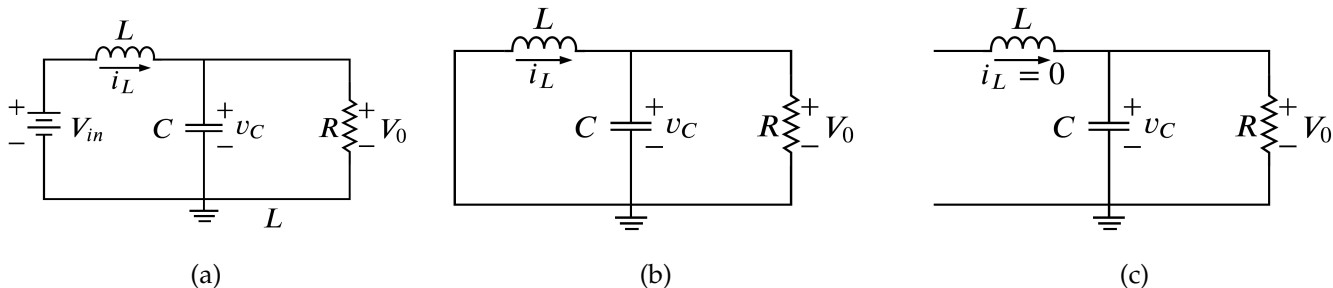

(a)　　　　　　　　　　　　　　　　(b)　　　　　　　　　　　　　　　　(c)

**Figure 2.** Topology of the equivalent circuits for a: (**a**) continuous conduction mode, (**b**) continuous conduction mode with the diode in forward bias, and (**c**) discontinuous conduction mode with the diode in reverse bias.

Under the assumption of an ideal switch $S$ and diode $D$, the Buck converter is a discontinuous system that can be described using a piecewise smooth mathematical model, whose vector field presents discontinuities. The mathematical model for the circuit working in a CCM, as in Figure 2a, is given by:

$$\begin{bmatrix} \frac{dv_C}{dt} \\ \frac{di_L}{dt} \end{bmatrix} = \begin{bmatrix} -\frac{1}{RC} & \frac{1}{C} \\ -\frac{1}{L} & 0 \end{bmatrix} \begin{bmatrix} v_C \\ i_L \end{bmatrix} + \begin{bmatrix} 0 \\ \frac{V_{in}}{L} \end{bmatrix} \quad for \quad S = 1 \ \& \ i_L > 0 \qquad (1)$$

where $S = 1$ denotes the CCM for the switch $S$ in state ON, and the positive current in the inductor.

On the other hand, the mathematical model when the circuit operates in a CCM as in Figure 2b, corresponding to $u = 0$ in (1), is described as:

$$\begin{bmatrix} \frac{dv_C}{dt} \\ \frac{di_L}{dt} \end{bmatrix} = \begin{bmatrix} -\frac{1}{RC} & \frac{1}{C} \\ -\frac{1}{L} & 0 \end{bmatrix} \begin{bmatrix} v_C \\ i_L \end{bmatrix} \quad for \quad S = 0 \ \& \ i_L > 0 \tag{2}$$

Finally, the mathematical model when the circuit works in a DCM and the diode is not conducting, as shown in Figure 2c, is described as:

$$\frac{dv_C}{dt} = -\frac{1}{RC} v_C \quad for \quad S = 0 \ \& \ i_L = 0 \tag{3}$$

Hence, as the reader can realize, the operation of the Buck converter is mainly driven by the status of the switch $S$ and system parameters, e.g., load parameter $R$, leading to a time response resultant from the combination of the three operative modes described in Figure 2. Thus, the response would eventually combine the three topologies given by Equations (1)–(3).

### 2.2. Buck Converter Voltage Control with Constant Hysteresis

Assuming that $V_{ref}$ is the voltage reference for the output of Buck power converter to feed an electric resistive load $R$, the control problem is to find in real time the right switching pattern $u \in \{0, 1\}$ to drive the switch $S$ and ensure that the voltage output $v_C$ reaches the target voltage $V_{ref}$, ensuring a suitable voltage supply to the electronic load $R$, even in the presence of disturbances.

The closed-loop error to solve the voltage control problem is defined as:

$$e = V_{ref} - v_s, \tag{4}$$

where $v_s = a.v_C$ is the output voltage measured by a sensor, with $a > 0$ as a positive sensor gain.

In this study, instead of choosing a linear controller combined with a PWM signal, we are interested in applying a discontinuous controller to assess the Buck converter response and performance. Thus, the control pattern $u$ is computed using a hysteresis logic $H(e, \xi)$, where $e$ is the error and $\xi$ is the hysteresis amplitude parameter.

$$u = \mathrm{H}(e, \xi), \tag{5}$$

where H is the hysteresis function, whose discontinuous behavior and memory feature are shown in Figure 3. Here, Figure 3a describes the nonlinear law in the error space, illustrating how the control transition occurs from 0 to 1 and vice versa. Note that such a control logic is determined for the hysteresis amplitude parameter, namely $\xi$. For numerical simulations purposes, Figure 3b represents the hysteresis control logic as a state machine, illustrating the memory property and the error dynamics conditions to make the control state transitions, defined as $u = 1$ and $u = 0$.

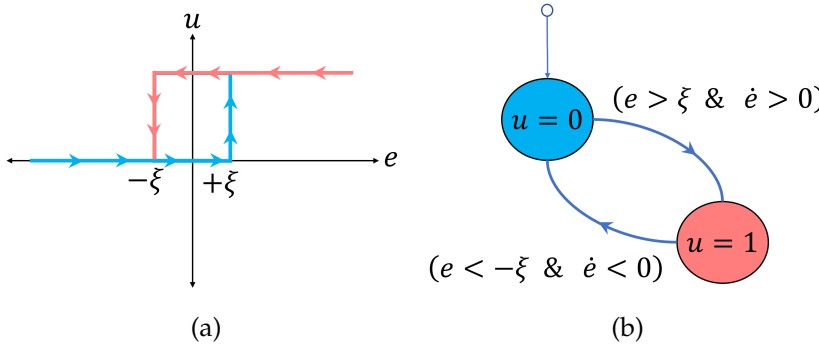

(a)                                        (b)

**Figure 3.** Hysteresis function: (**a**) hysteresis loop as a function of tracking error; (**b**) state machine to describe the control transition as a function of error dynamics, hysteresis amplitude parameter and control state.

Schematics of the closed-loop control for the Buck converter using the hysteresis control is shown in Figure 4. The driver module corresponds to the interface gate circuit between the electronic controller and the power stage, whose function is to enable/disable the power transistor acting as the switch $S$. More in detail, the voltage output $v_C$ in the Buck converter load $R$ is measured using a resistive divider as sensor, which provides the feedback signal $v_R = a.v_C$, to the voltage hysteresis controller. The tracking error $e$ is computed by comparing the desired voltage $V_{ref}$ to the actual measured voltage $v_s$. Then, based on the error information and hysteresis amplitude $\xi$ [V], the hysteresis control law chooses the right control state value, $u = 0$ or $u = 1$, to trigger the power transistor in the Buck converter.

Considering that a logic signal $u$ cannot enable/disable the power transistor, a suitable gate interface circuit is included in the module driver, as shown in Figure 4. Note that the scalar gain $a$ is the sensor gain used to measure the voltage output, and through this sensor gain, $a$ is possible to scale the Buck converter output using the same control architecture, allowing a wider range to be regulateds for voltage output.

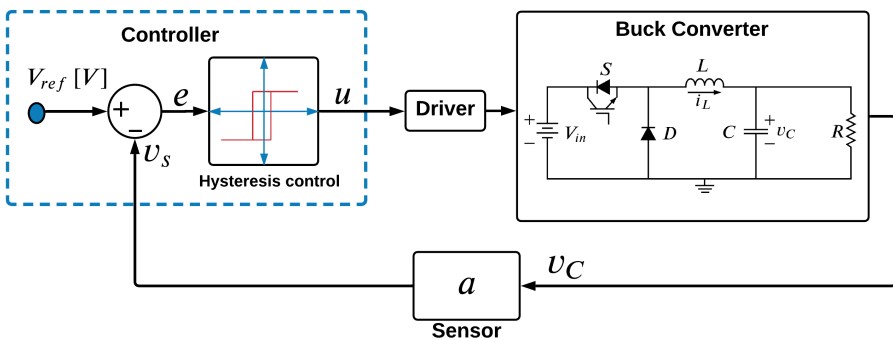

**Figure 4.** Schematics of the closed-loop control system for the Buck converter using a constant hysteresis control.

The following sections are dedicated to theoretically and experimentally analyzing the control performance in both transient and steady-state operations. Moreover, considering the hysteresis amplitude parameter, two control operations are analyzed: the first corresponds to the case with zero hysteresis control, and the second with constant hysteresis control.

### 2.3. Physical Considerations

The hysteresis control must drive the Buck converter switch $S$ to regulate the voltage output and meet the load application requirement. Figure 4 shows the Buck converter working with a closed-loop controller, leading to the study of a simple and interesting automatic control system. This control is used to regulate the output voltage $v_c$ at a desired scaled reference voltage $\frac{V_{ref}}{a}$ by setting the switch state in real time, independent of external disturbances such as load variations or changes in internal system parameters. Thus, the system is configured in a closed-loop circuit to self-regulate its behavior while rejecting eventual disturbances.

The constant hysteresis control includes the particular case of the zero hysteresis control [38], which corresponds to the well known bang–bang [39] or on–off control reported in the previous literature. Thus, in this paper, both zero hysteresis and constant hysteresis controls are analyzed.

### 2.4. System Parameters for Simulation

Numerical simulations of the Buck converter are performed considering the model parameters shown in Table 1. Numerical solutions are obtained using a theoretical solution obtained for the system of Equations (1)–(3).

**Table 1.** Buck converter system parameter.

| Parameter | Description | Value |
|-----------|-------------|-------|
| $V_{in}$ | Input voltage | 20 V |
| $V_{ref}$ | Reference voltage | 15 V |
| $R$ | Resistance | 22 $\Omega$ |
| $C$ | Capacitance | 1000 μf |
| $L$ | Inductance | 7 mH |
| $X_o$ | Initial conditions | 0 V ; 0 A |
| $\xi$ | Hysteresis band | 0 V |
| $dt$ | Time discrete steps | $1 \times 10^{-6}$ s |
| $t_o$ | Initial simulation time | 0.0 s |
| $t_f$ | Final simulation time | 0.4 s |

*2.5. Model Simulations and Numerical Methods*

To improve the accuracy of numerical simulations, the numerical bisection method described in Appendix A is implemented to handle the boundary crossing, detecting the switching event and capturing the initial conditions for the following state trajectory, when the system switches between solutions. Thus, when the trajectory crosses the boundary surface $v_C = V_{ref}$, the algorithm is enabled to find the right time value when the crossing takes place.

For instance, Figure 5a shows the evolution of the resultant switched trajectory when simulations are run using constant sampling time. Here, the trajectory is enlarged to show a common numerical simulation issue for piece-wise smooth systems due to the switching boundary conditions. Note that the trajectory diverges to greater oscillations, and the hypothesis of the unstable point is false due to this behavior being induced by error propagation during numerical simulations. Indeed, the bisection numerical method is applied at each switching of the system to manage the system transition properly and ensure the proper computation of trajectory boundary conditions for the next piece of the solution. Then, numerical error propagation is avoided between solution pieces and its transitions, leading to a more accurate complete simulation of system dynamics. Therefore, the application of this numerical algorithm allows improving the time-continuous numerical simulation, leading to a more accurate trajectory, as shown in Figure 5b. Hence, numerical simulations are reliable to obtain the correct hypothesis about the system dynamics.

For the sake of completeness, Figure 6a,b shows the convergence and evolution of the bisection algorithm for one event detection to find the numerical solution for the time value and voltage value, as the time converges to the solution. Note that derivation of the time solution for the Buck converter evolution is out of scope in this paper.

*2.6. Nonlinear Dynamics for the Zero Hysteresis Control*

Schematics of the feedback control applied to the Buck converter with a zero-hysteresis band are shown in Figure 7. Hence, the closed-loop system presents two strong non-linearities: the first is inherent in the Buck converter due to its variable structure characteristic of working with a diode $D$ and a switch $S$; and the second is added by the nonlinear nature of the hysteresis control. In particular, Figure 7a shows the block diagram for the zero hysteresis control with $\xi = 0$, while Figure 7b shows the zero hysteresis control as a function of the output voltage $v_C$ (for $a = 1$). The control generates a binary signal defined as $u = 0$ and $u = 1$ depending on the error sign. Thus, when the output voltage $v_C$ is under the reference voltage $V_{ref}$, the system must be energized with a signal $u = 1$. When the controller increases, the output voltage overcomes the reference voltage $V_{ref}$, the power transistor must be turned off using a signal $u = 0$. The system will be maintained indefinitely, making the transition of states, increasing and decreasing the output voltage, while switching states from ON to OFF and vice versa. Ideal conditions will lead to an infinite switching frequency with zero ripple; however, the experimental output voltage presents ripples, as shown in following section, due to physical constraints and the dynamic

behavior of electronic devices, which can actuate at high (finite) switching frequency with joule power losses.

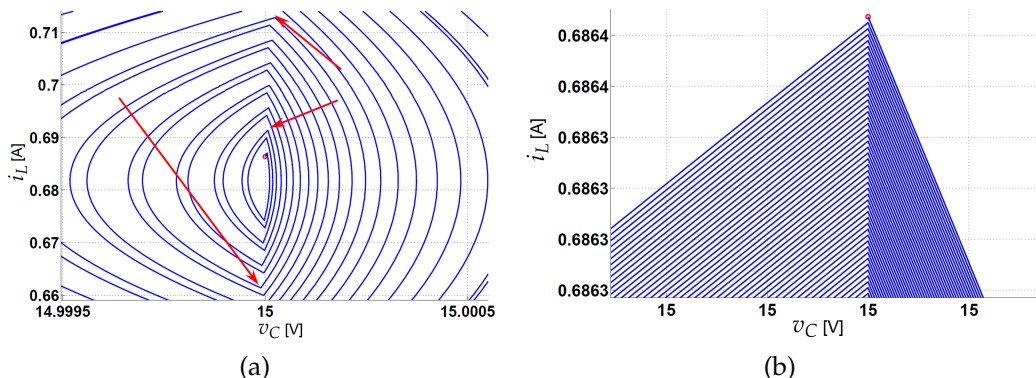

**Figure 5.** Application of the numerical method to solve an accuracy error on switching boundary: (**a**) state trajectory without the numerical algorithm and (**b**) state trajectory with the numerical algorithm.

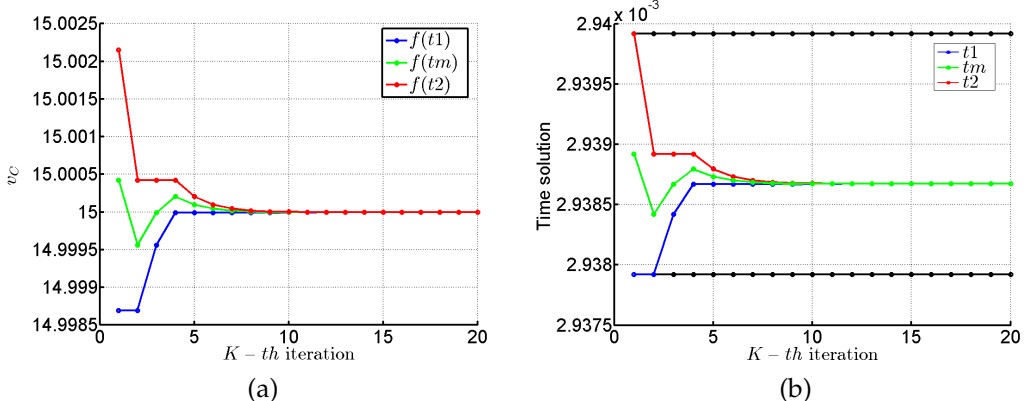

**Figure 6.** Bisection numerical method results: one example to show evolution and convergence of numerical solution: (**a**) boltage value convergence to $v_C = 15$ and (**b**) time solution evolution.

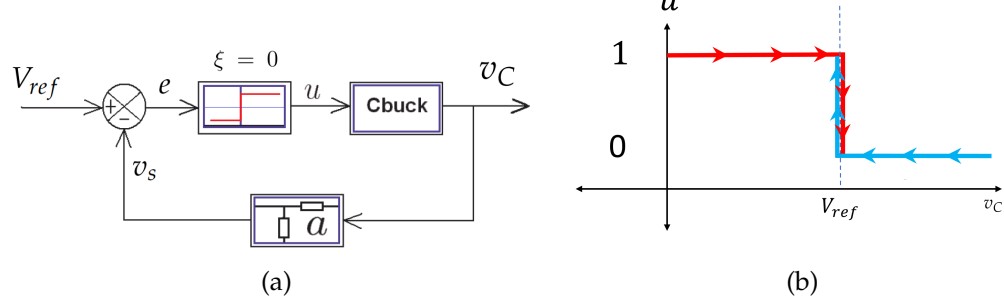

**Figure 7.** Hysteresis control applied to the Buck converter (**a**) block diagram with the zero hysteresis control and (**b**) switching state for the zero hysteresis control ($\xi = 0$ V).

### 2.6.1. Steady- and Transient-State Simulations

Simulations are performed to obtain the dynamic behavior in various operating conditions when the zero-hysteresis strategy controls the system. The simulation is performed with the parameters of the circuit obtained in the laboratory to contrast the results with the implemented circuit. This test aims to identify the dynamic behavior of the system during the steady- and transient-state operation. The result of the simulation is shown in Figure 8, where Figure 8a shows the phase diagram and Figure 8b shows the temporal evolution diagram.

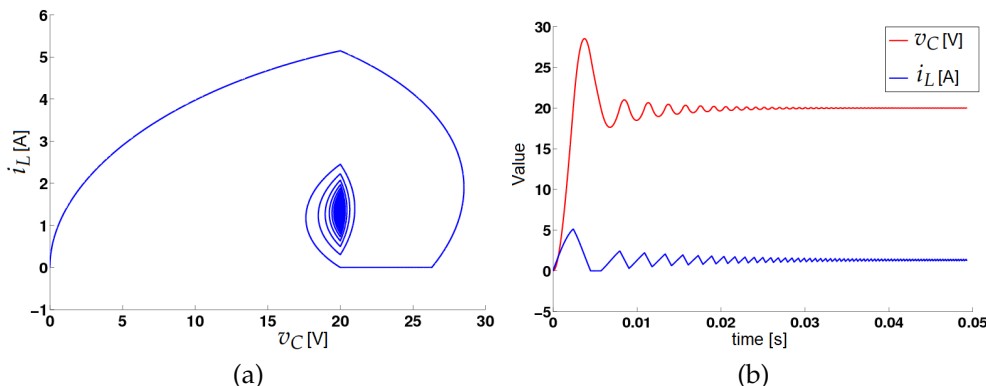

**Figure 8.** Dynamic behavior in steady- and transient-state operations: (**a**) phase diagram and (**b**) temporal evolution.

The system presents during the transient state a large overshoot in a short period of time associated with the natural oscillation frequency of the *LC* filter. This phenomenon is not desirable for applications, and the signal must be regulated by eliminating the overshoot present during the transient state. In the steady-state operation, the system converges quickly to an equilibrium point, showing good system regulation. Note that the undesired overshoot found in the Buck converter with zero hysteresis controller, can be improved by adding a reference governor [40] to the set point, avoiding sudden transitions of reference signal $V_{ref}$.

### 2.6.2. Regulation for Different Voltage Set Points

This study aims at showing the regulation capability when the chosen required voltage is low (5 V) or high (25 V) with respect to the input voltage (30 V), implying the switching of the system in two opposite regions in the phase portrait, where trajectories are switched with different vector field intensity, according to the switch S state, combining two topologies for $u = 1$ and for $u = 0$ in CCM. From simulation results, as shown in Figures 9a,b, it is clear that transient evolution of the resultant trajectory (merged orbit) presents an asymmetric and damped ripple due to the non-smooth nature and intensity of the vector field. Here, Figure 9a shows the simulation for a low reference voltage $V_{ref} = 5$ V, and Figure 9b shows the simulation for a high reference voltage $V_{ref} = 25$ V.

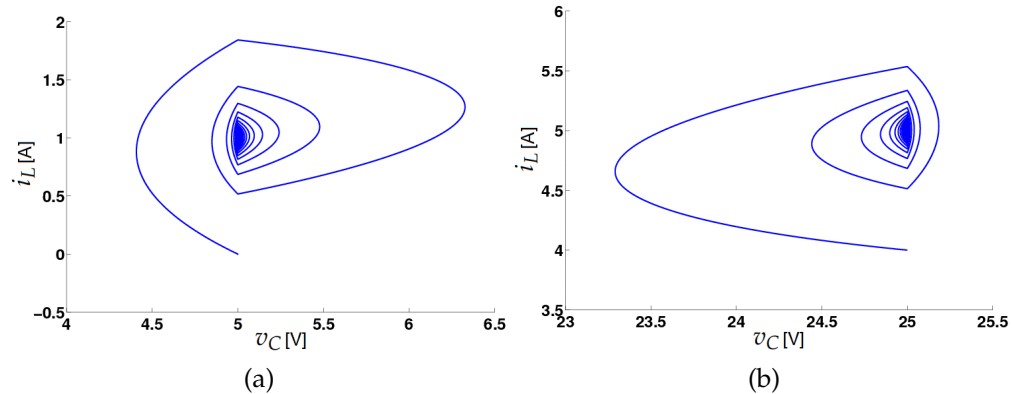

**Figure 9.** Asymmetry in the phase plane for (**a**) low reference voltage $V_{ref} = 5$ V and (**b**) high reference voltage $V_{ref} = 25$ V.

This nonlinear event generates asymmetry in the ripple of the voltage signal and can affect the DC voltage regulation (mean value) during the transient operation. The time evolution is also shown in Figure 10a,b, where it is evident that the greater overshoot and ripple occurs, when the set regulation point is shown to be much lower with respect to

supply voltage $V_{in}$. From an application viewpoint, this observation suggests, for example, that the ratio $\frac{1}{4} < \frac{V_{ref}}{V_{in}} < 1$, should be taken into account to avoid undesired overshoots and ripples. Of course, more sophisticated control strategies for power converters can better address the overshoot problem and ripple asymmetry. Note that considering the reference voltage $V_{ref} = \frac{V_{in}}{2}$ for vector fields with equivalent intensity, the resultant voltage trajectory exhibits an equal distributed ripple, as shown in Figure 11a.

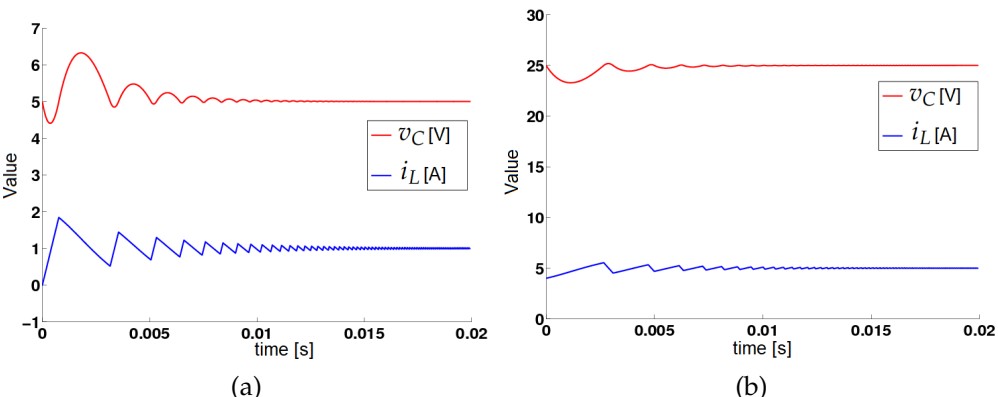

(a)                                             (b)

**Figure 10.** Temporal dynamics of (**a**) low reference voltage $V_{ref} = 5$ V and (**b**) high reference voltage $V_{ref} = 25$ V.

During the transient-state operation, the hysteresis control generates a variable commutation pattern, and the closed-loop system converges to the resultant equilibrium point driven by the set point $V_{ref}$ as given by Equation (6). Then, the switching frequency increases as shown in Figure 11b, where simulation allows inferring that the system has a fixed point with an infinite switching frequency.

$$\begin{bmatrix} v_C^* \\ i_L^* \end{bmatrix} = \begin{bmatrix} V_{ref} \\ \frac{V_{ref}}{R} \end{bmatrix} \tag{6}$$

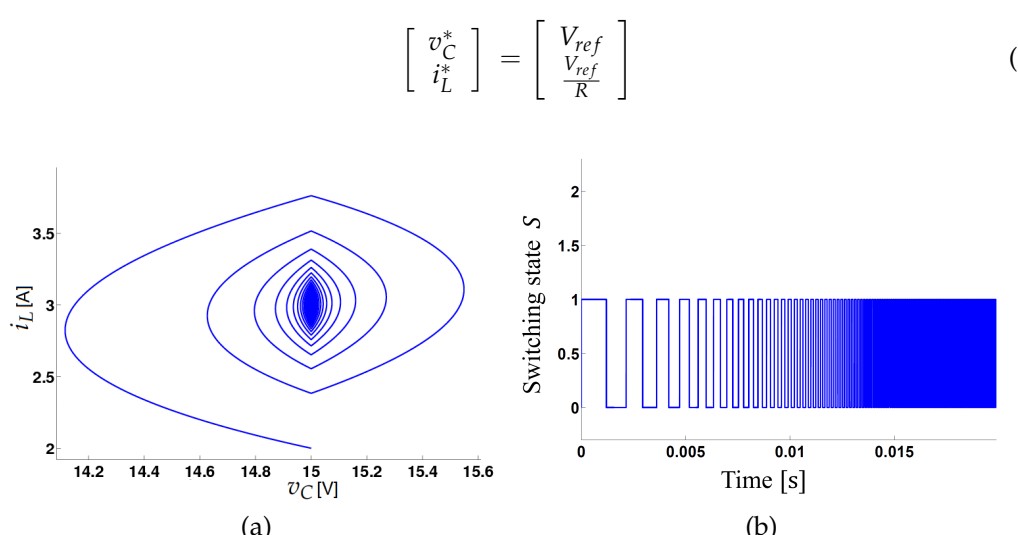

(a)                                             (b)

**Figure 11.** Dynamics when considering symmetric vector fields with $V_{ref} = 15$ V: (**a**) phase diagram and (**b**) temporal evolution of switching pattern for switch $S$.

### 2.7. Nonlinear Dynamics for the Constant Hysteresis Control

As expected, the application of a hysteresis control to regulate the Buck converter voltage output according to schematics in Figure 12 allows controlling the output voltage with a reduced switching frequency and induced oscillations around the required voltage $V_{ref}$. Thus, Figures 12a,b show the diagram with the constant and hysteresis control and the switching state for the constant hysteresis control, respectively. Such induced oscillations can be characterized as stable $1T-$periodic orbits that can evolve in CCM mode (see Figure 13a)

with $i_L > 0$, involving two topologies (Equations (1) and (2)). Furthermore, it can evolve in DCM mode with $i_L \geq 0$ (see Figure 13b) combining three topologies (Equations (1)–(3)). Both types are observed via numerical simulations and verified experimentally, as shown below.

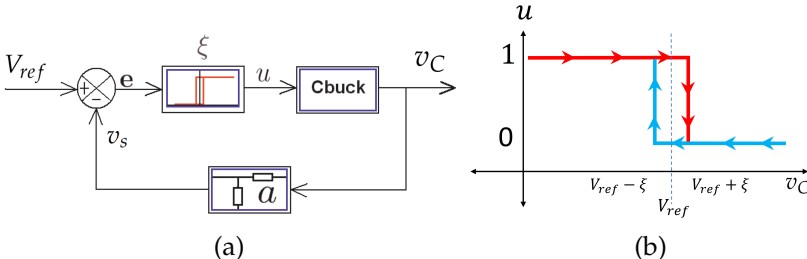

(a)        (b)

**Figure 12.** Hysteresis control applied to the Buck converter (**a**) block diagram with the constant hysteresis control and (**b**) switching state for the constant hysteresis control ($\xi > 0$).

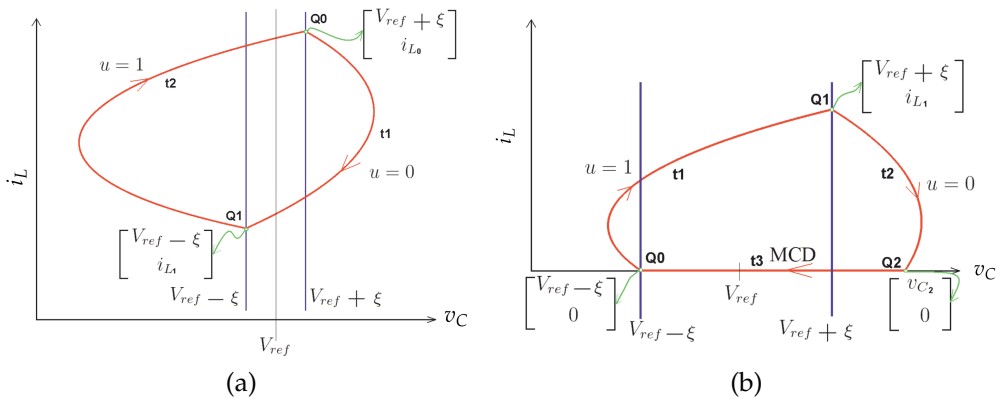

(a)        (b)

**Figure 13.** $1T$-periodic orbits with the constant hysteresis control: (**a**) $1T$-periodic orbit in CCM mode ($i_L > 0$), combining two solution pieces for $S = 1$ and $S = 0$; and (**b**) $1T$-periodic orbit in DCM ($i_L \geq 0$), combining three solution pieces for $S = 1$, $S = 0$ in CCM and for $i_L = 0$.

The time evolution for a typical trajectory for the Buck converter driven with constant hysteresis is shown in Figure 14, where Figure 14a shows the phase portrait and Figure 14b shows the time evolution. For this simulation case, the system converges to the stable $1T$-periodic orbit in CCM, of course with a finite switching frequency. Note that, similar to the case of zero hysteresis control, the Buck converter presents a strong transient behavior and asymmetry in the ripple when different voltage targets are required, as shown in Figure 15. Here, the simulation includes two trajectories with different initial conditions, both inner and outer, with respect to the limit cycle, to show the trajectory convergence to the stable $1T$-periodic orbit in CCM.

Figures 15a,b shows that switching frequency and ripple can vary as a function of the control parameter hysteresis amplitude $\xi$. Moreover, they can also vary as a function of other system parameters and a required voltage $V_{ref}$. As an example, to illustrate the effect of hysteresis parameter on the output voltage, Figure 16 shows the phase portrait for different values of $\xi$, where it is evident the presence of a Grazing bifurcation at $\xi \approx 0.83$; whereby increasing $\xi$ the CCM $1T$-periodic orbit is transformed into a $1T$-periodic orbit in DCM with $i_L = 0$.

Moreover, Figure 17a shows the mean value $\mu$ and variance $\sigma^2$ as a function $\xi$ parameter, where it is clear how the mean value increases and is greater than the required voltage $V_{ref} = 12$ V, due to the ripple effect. Moreover, Figure 17b shows that the variation rate presents a discontinuity that is linked to the Grazing bifurcation point at $\xi \approx 0.83$, which means that the $1T$-periodic orbit in the CCM model turns into a $1T$-periodic orbit in the DCM with $i_L \geq = 0$.

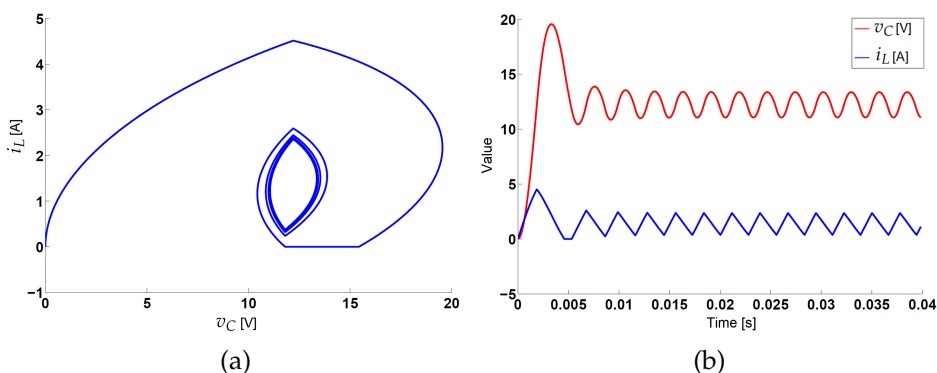

(a)

(b)

**Figure 14.** Voltage control using the constant hysteresis control for $\xi = 0.2$ V: (**a**) phase portrait and
(**b**) time evolution.

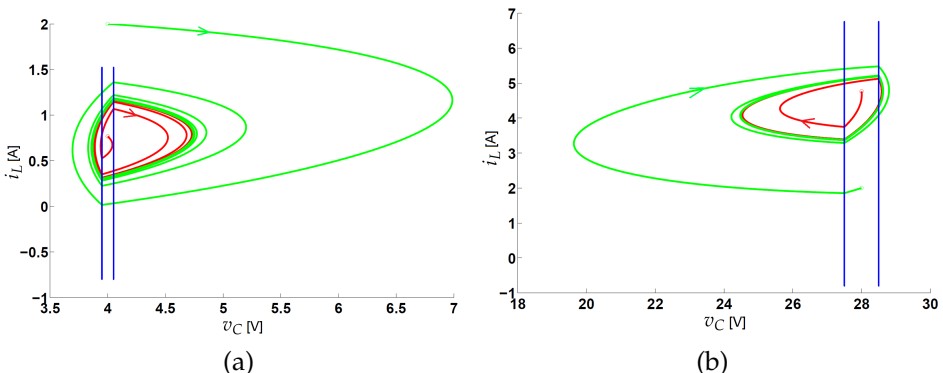

(a)

(b)

**Figure 15.** Voltage control using the constant hysteresis control for $\xi = 0.2$ V: (**a**) phase portrait and
(**b**) time evolution.

Ripple variations could be solved using an adaptive hysteresis control and obtaining
a symmetric regulation of voltage ripple due to system variations. This allows obtaining
similar ripple for a wide range of operative conditions.

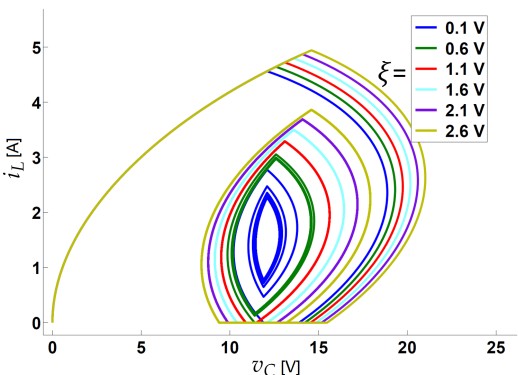

**Figure 16.** Phase portrait varying the hysteresis control parameter $\xi$. Increasing parameter $\xi$ a
**Grazing bifurcation** appears to transform the CCM $1T$-periodic orbit into a $1T$-periodic orbit in
DCM with $i_L \geq 0$.

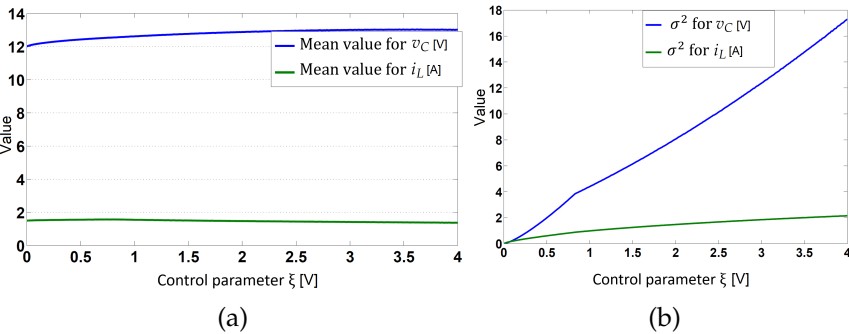

**Figure 17.** Regulation indexes mean and variance when hysteresis parameter $\zeta$ is varied: (**a**) mean values and (**b**) variance.

## 3. Experimental Results

### 3.1. Experimental Implementation of the Zero Hysteresis Control

The diode and the transistor are described as ideal switches in the mathematical model derived for the Buck converter. Furthermore, the resistive load, the inductor, and the capacitor are taken as pure impedance without considering the existence of internal parasitic impedance. Experimentally, the operating condition with zero hysteresis control cannot be obtained because electronic components such as MOSFET transistors and diodes have physical limitations inherent to their functioning. Hence, the time response, activation times and some physical parameters were not considered to avoid a complex model. Therefore, these approximations generate non-correspondence with model simulations in some operating conditions, as shown below. The parameters considered in the tests are presented in Table 2, where $a = 1/3$ is the gain of the sensor.

**Table 2.** Parameters of the Buck converter with the zero-hysteresis control for the experimental test.

| Parameter | Description | Value |
|:---:|:---:|:---:|
| $V_{in}$ | Input voltage | 30 V |
| $V_{ref}$ | Reference voltage | $--$ V |
| $a$ | Constant of the sensor | 1/3 |
| $R$ | Load resistance | $--$ Ω |
| $C$ | Capacitance | 299 μf |
| $L$ | Inductance | 10.6 mH |
| $R_{inL}$ | Resistance of the inductor | 0.4 Ω |

Figure 18 shows the experimental prototype and test bench, and Figure 19 shows the control circuit and the Buck converter used in the experimental prototype. The switching element corresponds to a MOSFET P-channel transistor (IRF9630). The control rule with the zero hysteresis strategy is implemented with analog electronics.

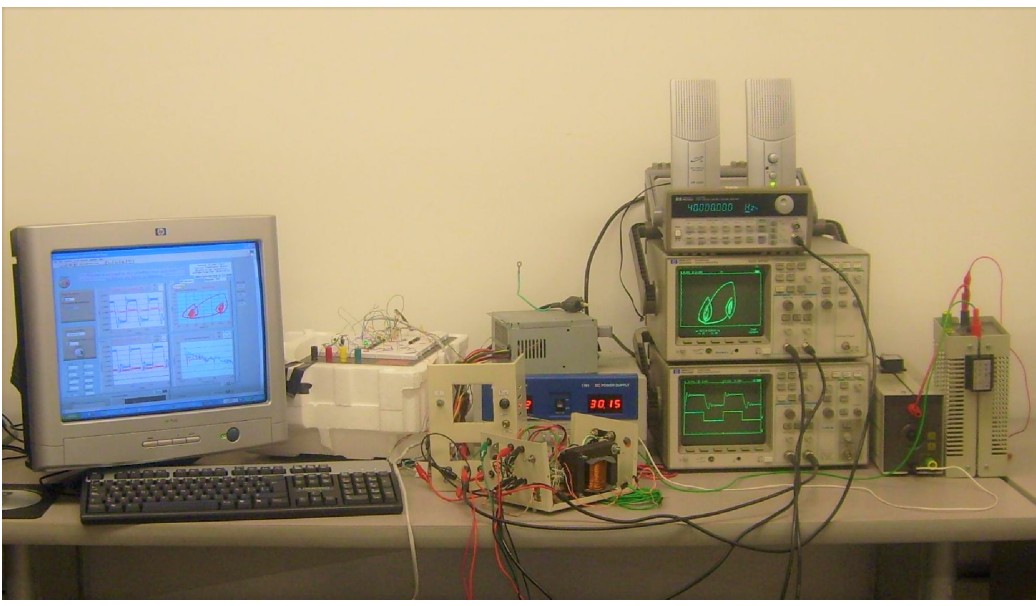

**Figure 18.** Test bench and experimental prototype for the Buck Converter with Hysteresis Control.

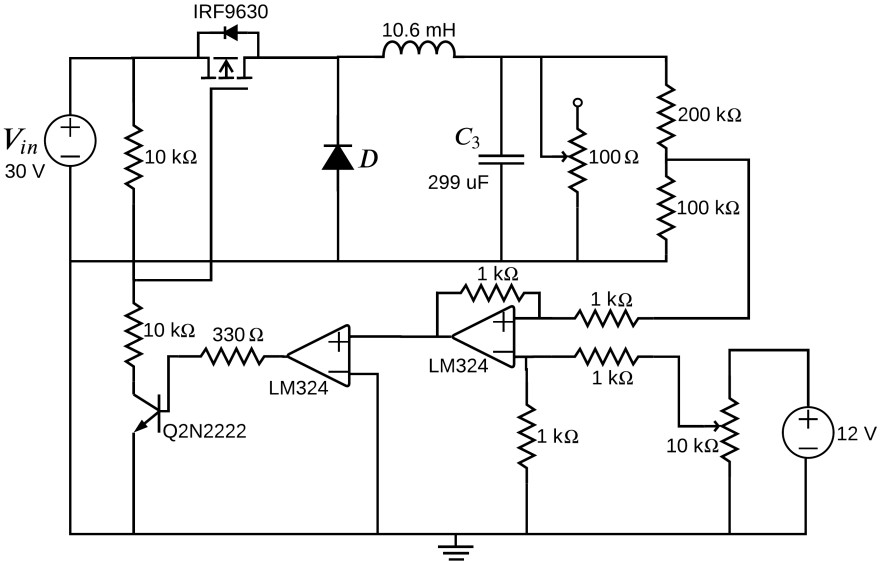

**Figure 19.** Electronic circuit of the Buck converter with zero hysteresis control.

Comparison of Experimental Observation with Simulations

A simulation was carried out with a load disturbance, and the trajectories are observed in Figure 20. Figure 20a shows the evolution of the trajectory in the phase plane for an initial load of 8 $\Omega$ and a final value of 4 $\Omega$. The attractor of the equilibrium point, after a transient behavior, tends to stay in the vertical line $v_c = 5$ V, thus obtaining the desired regulation. A temporal dynamic operation is presented when the load step occurs as illustrated in Figure 20b, generating the transient state and increasing the inductor current $i_L$, while the output voltage tends to regulate at the reference voltage $V_{ref} = 5$ V. This transient occurs because the system moves to a new energy state and this energy level transition causes a dynamic transient in the state variables. Thus, the control with zero hysteresis rejects the disturbances in the load and allows the desired output to be obtained.

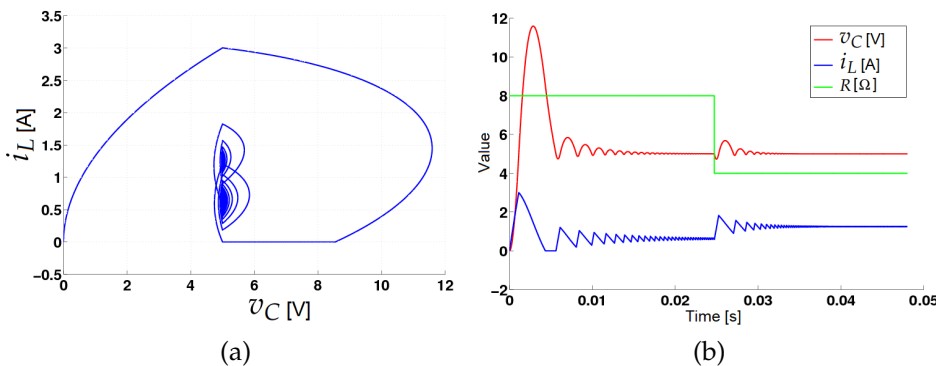

**Figure 20.** Dynamics with load perturbations: (**a**) phase plane and (**b**) temporal response.

*3.2. Steady- and Transient-State Operation*

A first result is obtained when the parameters are configured with the values $V_{ref} = 5$ V, the load $R = 8.5$ Ω, and the input voltage $V_{in} = 30$ V. Figure 21a shows the phase diagram of the real behavior of the Buck converter with the zero hysteresis control, and Figure 21b shows the temporal evolution obtained by simulation. In addition, Figure 21c shows the phase diagram obtained with the experimental test and Figure 21d shows the temporal evolution obtained with the experimental test.

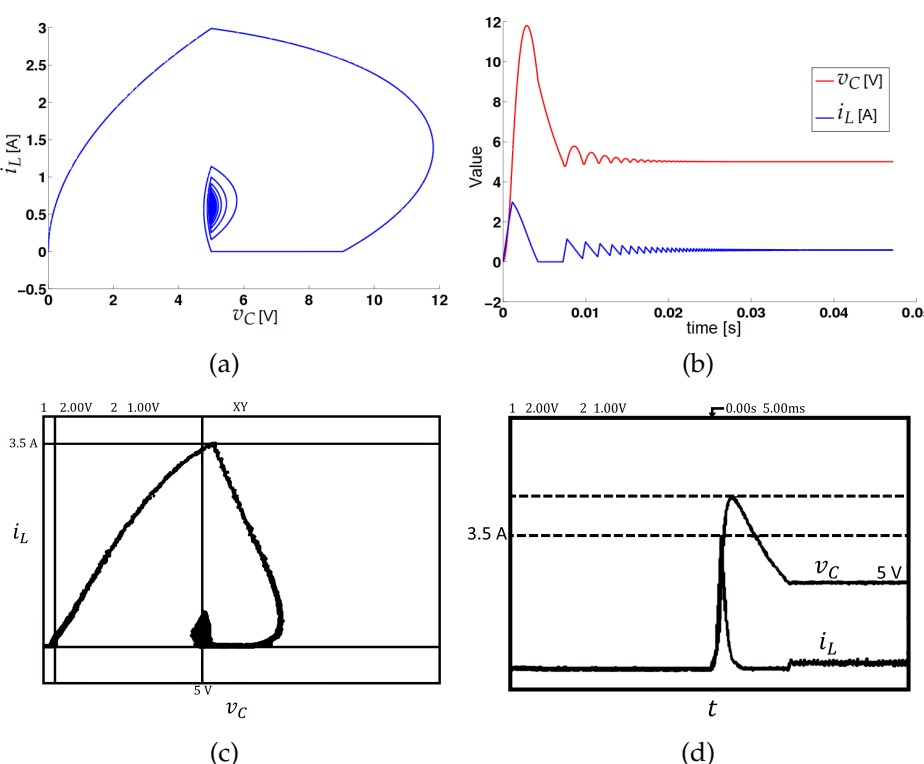

**Figure 21.** Numerical simulation vs. experimental results in the steady- and transient-state operations with $\xi = 0$ V and $v_c = 5$ V: (**a**) experimental phase diagram, (**b**) experimental temporal evolution, (**c**) theoretical phase diagram, and (**d**) theoretical temporal response.

The dynamic behavior obtained in the experimental test presents a similar qualitative and quantitative behavior in comparison with the responses obtained in the simulation test. In the experiment, the voltage presents an overshoot of 12 V and the commutation frequency in the steady-state operation corresponds to 2.5 kHz, showing the existence of a limit cycle as a consequence of the natural hysteresis of the system. This is a phenomenon presented when the current is lower than 1.5 A and the reference voltage is lower than 15 V. For other values outside this range, the system enters into chaos, where different

switching frequencies up to 50 kHz are presented for some parameters of the operation. In addition, these conditions are presented due to the high switching frequencies of the transistor, generating overheating and deterioration and reducing the life cycle.

### 3.2.1. Load Disturbance

In order to detail the response of the system, when the load is disturbed and to contrast the dynamic behavior obtained by simulation, the system is configured with the voltage reference $V_{ref} = 14.53$ V, a first load $R_1 = 31.8\ \Omega$, a second load $R_2 = 8.5\ \Omega$, and the output voltage $v_c = 14.35\ V$. For this purpose, the initial load is 31.8 $\Omega$ and changes to 8.5 $\Omega$. The phase and temporal evolution diagrams of the state variables are shown in Figure 22, where the transient phenomenon is plotted.

Transient behavior is presented under the regulation voltage when the load increases, which implies that the load is not subjected to voltages that exceed the regulation value. On the other hand, when an initial load of 8.5 $\Omega$ is disturbed to 31.8 $\Omega$, the voltage is above the regulation value, and this can be harmful to the load connected to the system. The disturbance of the load can be extended, in the case that there are several shunt loads connected to the system and some of them present a fault, which would mean an alteration in the quality of the wave for the other loads.

Figure 22 shows the numerical simulations and experimental results when the load increases and decreases. In this figure, ripple is shown in the state variables when the load is changed, which is related to the change in the commutation frequency of the switch when the load is varied. This ripple has a proportional dependence in relation to the load with which the system operates. A simulation is performed with the parameters corresponding to Table 1, to detail both disturbances: load increasing and load decreasing.

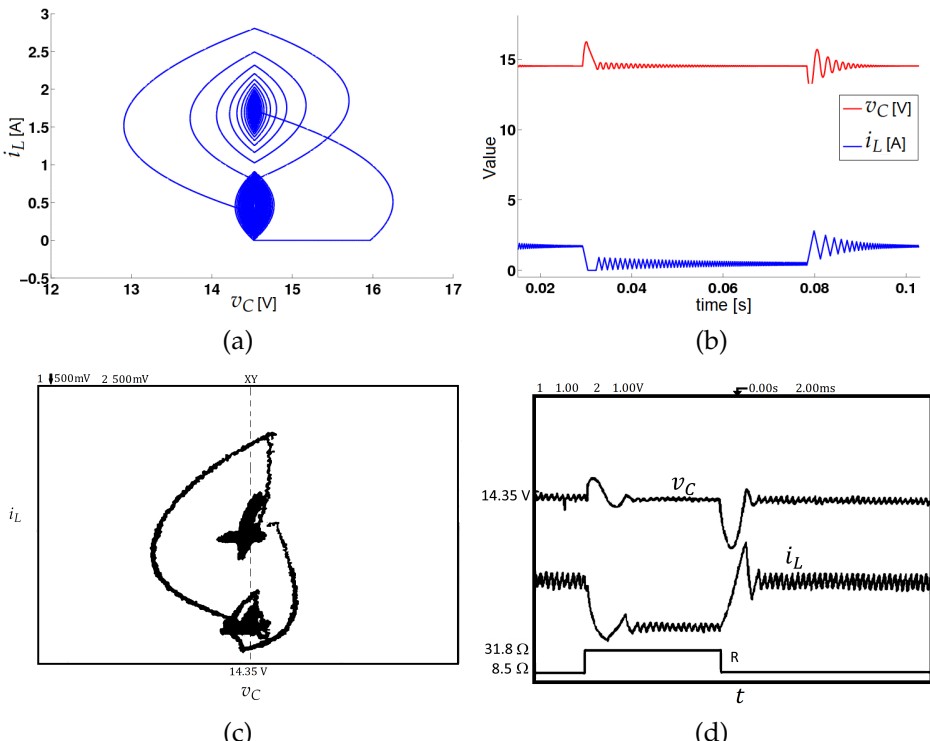

**Figure 22.** Comparison of simulation and experiment of the zero hysteresis control when the load increases and decreases, and the reference voltage is $V_{ref} = 14.53$ V for a (**a**) simulated phase diagram, (**b**) simulated temporal evolution, (**c**) experimental phase diagram, and (**d**) experimental temporal evolution.

### 3.2.2. Chaos with the Zero Hysteresis Control

Experimental testing showed that some operation conditions lead the system to present chaotic behavior. For example, to obtain this behavior, the circuit was configured with the following values $V_{ref} = 20$ V and $R = 25$ Ω. For this configuration, Figure 23a clearly shows the voltage output and current signals with variable ripple, and in Figure 23b, these same signals are shown but taking into account only their AC component. Such chaotic behavior has low variable ripple because it is not possible to implement a zero hysteresis band; in turn, a residual hysteresis takes place due to physical limitations and internal parasitic effects of electronic components. This undesired behavior can be enhanced by using more technological electronic components.

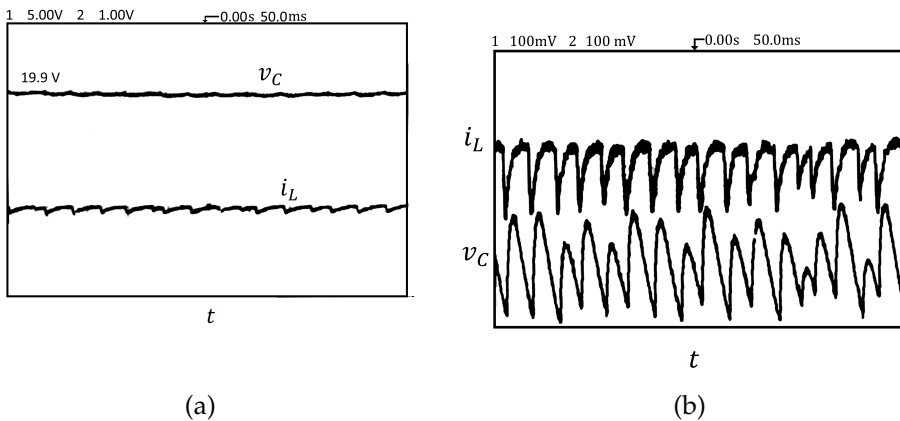

(a)                                             (b)

**Figure 23.** Experimental chaotic dynamics with the zero hysteresis control due to existence of residual hysteresis created by physical limitations and internal parasitic effects of electronic components: (**a**) DC voltage output and inductor currrent and (**b**) AC signals showing the variable ripple on voltage and current.

Another example of the chaotic evolution with a small ripple is shown in Figure 24a, using a greater target voltage $V_{ref} = 26$ V. Here, the phase diagram shows the voltage ripple. Certainly, the chaotic attractor evolves in a small region around $v_C = 26.3$ $V$, due to the controller response, with voltage variations of 300 $mV$ and current variations of 200 $mA$. Figure 24b shows the voltage signal $v_c$ and the gate voltage control to enable the MOSFET transistor labeled as pulse control $Pc$, which also shows how the energizing time or pulse width presents variations, in turn creating an asymmetric ripple on state variables.

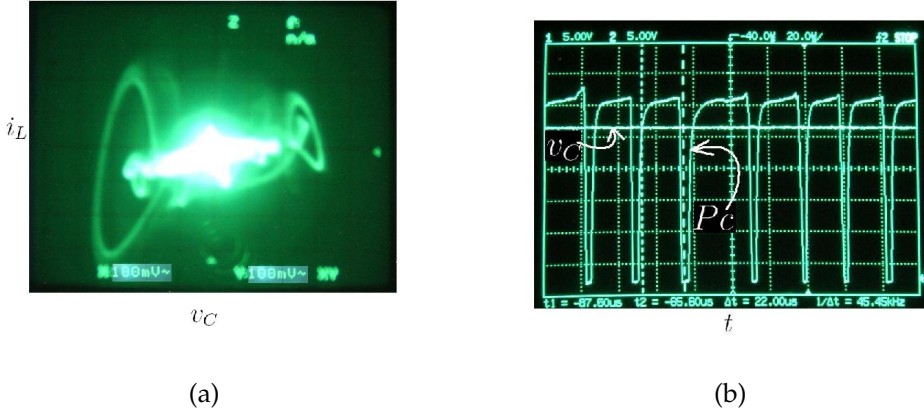

(a)                                             (b)

**Figure 24.** Experimental dynamic behavior with residual zero hysteresis control $\xi \simeq 0$ V, plotting (**a**) phase diagram and (**b**) temporal evolution.

Concerning the zero hysteresis control for reference voltages lower than 10 V, there are acceptable commutation frequencies as shown previously in the order of 2.5 kHz. However,

when the system operates for desired regulation values greater than 15 V, e.g., $V_{ref}$ = 26 V, a chaotic regime occurs, as shown in Figure 24a. Furthermore, it is shown that for the system with the zero hysteresis control, a chaotic behavior takes place with large commutation frequencies, for this case $f$ = 45.45 kHz as shown in Figure 24b. This high switching is undesired because the MOSFET transistor can be overhead due to energy losses and can be damaged. This behavior is not observed via simulations due to the fact the model assumes ideal physical behavior for system parameters and electronic components.

Below, some examples of real orbits in CCM are presented created with the following parameters: $R = 5.5\ \Omega$ and $V_{ref}$ = 12 V. Figure 25a,b shows the results. Furthermore, Figure 25c,d shows the real dynamics obtained from the experimental prototype. In addition, it is observed that the overshoot obtained with the simulation is 16.97% and with the experiment is 15.94%, with a regulation error less than 5%.

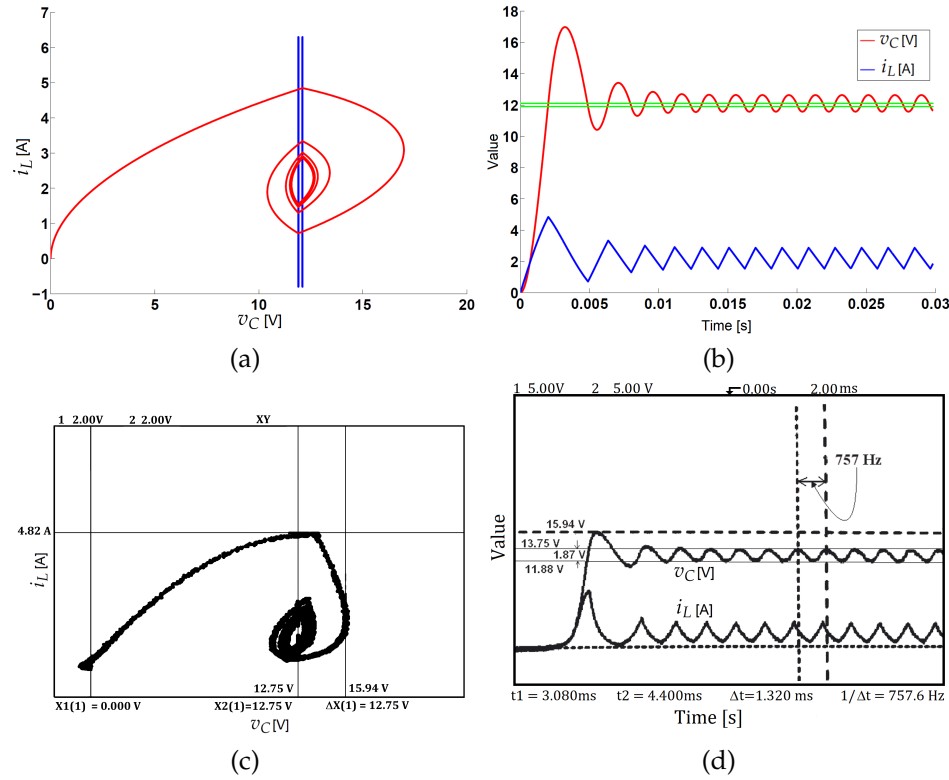

**Figure 25.** Experimental validation of voltage control using the constant hysteresis control for $\xi = 0.2$ V, converging to 1$T$-periodic orbit in CCM: (**a**) phase portrait, (**b**) time evolution for simulation results, (**c**) phase portrait, and (**d**) time evolution for experimental results.

### 3.3. Experimental Validation of the Constant Hysteresis Control

In an electronic converter, the load, the output voltage, and the hysteresis band have to be varied in order to make the system operate in DCM. Thus, a simulation is carried out by changing these parameters in order to detail the dynamics and validate the simulation with the real behavior of the system. The values of the parameters used in the test are $V_{ref}$ = 20 V and $R = 79.5\ \Omega$.

The dynamics illustrated in Figure 26a,b are obtained through simulations, in which the system slides by $i_L = 0$ A, because the diode and the transistor are inactive. In DCM, the voltage across the capacitor is reduced until it reaches the lower band of the hysteresis activating the switch, and then the inductor is energized, as reflected in the increase in current $i_L$. In CCM, the capacitor starts to be charged, generating an increase in the output voltage until it reaches the upper hysteresis band again.

Figure 26c,d shows the real dynamics captured with the experimental prototype. From the qualitative point of view, the similarity is acceptable. However, a detailed comparison

obtains that the value between the model and the real dynamics of the system presents some inconsistencies, especially during the time that the current remains at zero.

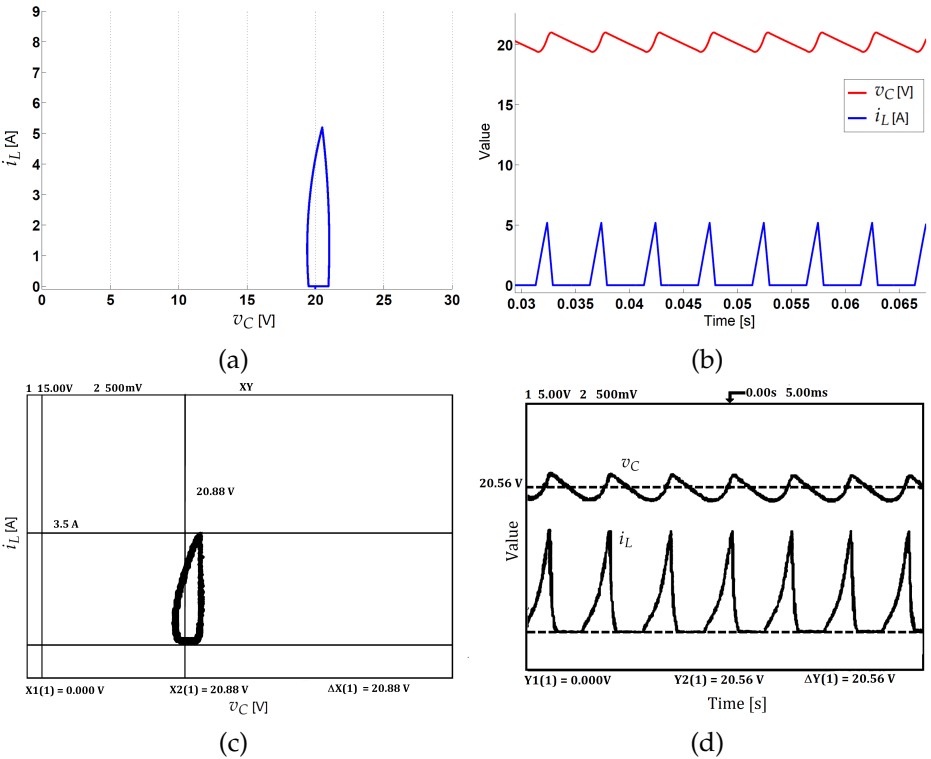

**Figure 26.** Experimental validation of voltage control using the constant hysteresis control for $\xi = 1$ V, converging to $1T$-periodic orbit in DCM: (**a**) phase portrait, (**b**) time evolution for simulation results, (**c**) phase portrait, and (**d**) time evolution for experimental results.

It is recommended to use a small hysteresis to obtain good tracking performance while ensuring a low switching frequency (2.5 kHz), as shown in Figure 27. In this case, the experimental prototype is configured to regulate the output voltage to 12 $V$. by imposing the target $V_{ref} = 12 \, V$. Note that the commutation frequency of the switch during the steady-state operation with the zero hysteresis control is around $fc = 2.5$ kHz, as shown in Figure 27. Because of the alteration of the operating conditions, the vector field induces a different type of ripple and the switching frequency tends to change. Indeed, the experimental test showed that the switching frequency is not constant for high reference values approaching the supply source $E$, leading to small cycles responsible for output ripple.

Note that the obtained results allow new learning regarding practical considerations and lessons to continue designing and improving power converters. For example, the undesired overshoot found in the Buck converter with hysteresis controller can be improved by adding a reference governor [40] to avoid sudden transitions in the set point. This improvement, combined with more advanced control techniques, and a better experimental prototype will lead to a better-controlled DC–DC power converter and it will be explored in future research.

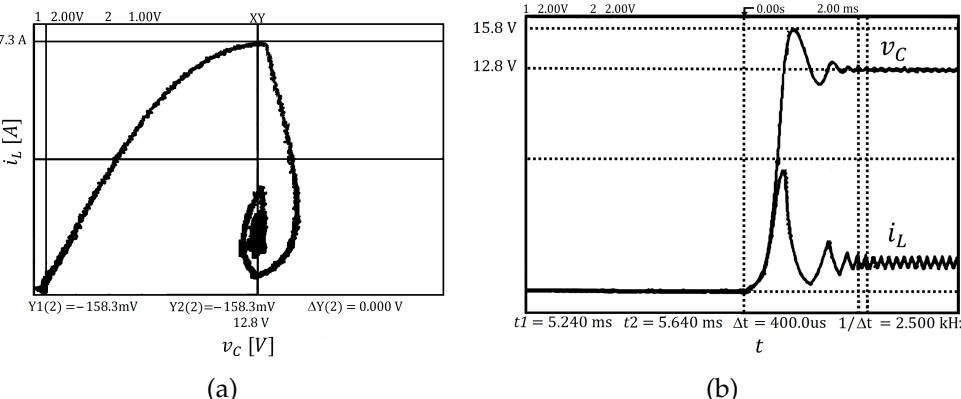

**Figure 27.** Experimental validation of voltage control using the constant hysteresis control with low value, e.g., $\xi$ = 0.05 V, and required voltage is $V_{ref}$ = 12 $V$: (**a**) phase portrait and (**b**) time evolution.

## 4. Conclusions

This paper presented the mathematical modeling and experimental implementation of a Buck converter with hysteresis control. Mathematical model description, the hysteresis control, and the experimental implementation using analog electronic devices were presented.

Two control configurations were explored—zero hysteresis and constant hysteresis—and numerical simulations were implemented using a bisection method to properly handle the initial condition on the switching boundaries to drive the model solution transitions.

The Buck converter with the zero hysteresis control presents a stable fixed point with the implication of an infinite commutation frequency, phenomena that cannot be validated experimentally due to physical constraints of electronic components, parasitic resistance, and internal dynamics. Although the experimental test was carried out with available commercial components, the results such as dynamic evolution, the voltage, and current ripples would be similar to those obtained with currently used components. Furthermore, fast switching components (transistor, diode, and operational amplifiers) can lead to obtaining a better regulation with low steady-state error and low-size LC filter.

A constant hysteresis control was successfully applied and validated experimentally to mitigate the high switching and joule loss issues caused by the zero hysteresis control. Some $1T$-periodic orbits were also predicted and validated experimentally. The output ripple is proportional to the hysteresis band amplitude parameter, while the switching frequency is inversely proportional, leading to a tradeoff when tuning the control parameter $\xi$ to choose appropriately between output voltage ripple and switching frequency. Some working conditions varying the target voltage lead to an output voltage with ripple polarization.

**Author Contributions:** Conceptualization, investigation, methodology, and software, C.I.H.V. Formal analysis, writing—review and editing, C.I.H.V., J.E.C.-B. and F.E.H.V. All authors have read and agreed to the published version of the manuscript.

**Funding:** This research received no external funding.

**Data Availability Statement:** Not applicable.

**Acknowledgments:** The authors would like to thank the Universidad Nacional de Colombia Sede Manizales for the support for conducting this project during MSc. studies of Carlos I. Hoyos Velasco, who contributed to manuscript editing as a research fellow at IM-CNR, of Naples, Italy. The work of Fredy E. Hoyos and John E. Candelo-Becerra was supported by Universidad Nacional de Colombia, Sede Medellín.

**Conflicts of Interest:** The authors declare no conflict of interest.

## Appendix A. Bisection Numerical Method

The bisection method is a procedure to find a solution numerically, namely a root in mathematical applications. Given a monotonic function $f(t)$ for $t \in [Ti, Tf]$, the problem is to numerically find the time solution $t = tr \in [Ti, Tf]$ such that $f(tr) = V_{ref}$, as described in Figure A1.

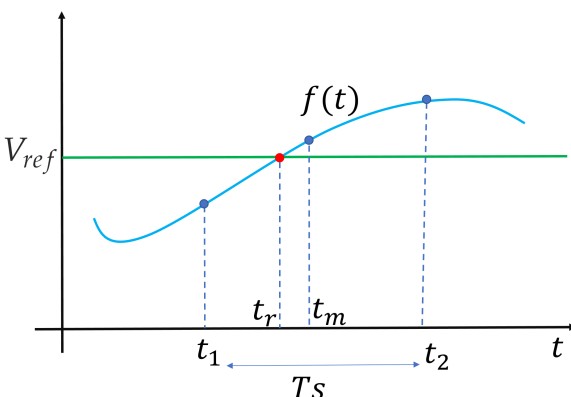

**Figure A1.** Graphical description for bisection numerical method.

To reach the existent solution in the time interval $[t1, t2] = [Ti, Tf]$, the algorithm at each $k - th$ iteration computes the time mean value $tm = 0.5(t1 + t2)$, bisects the time interval, and computes a new interval, where the root exists, to continue the search in the next iteration. The algorithm is iterated unil the convergence criteria $|(f(tm) - V_{ref})| < \epsilon$ is satisfied, leading to finding the root for the time solution, e.g., $tr = tm \in [t1, t2]$. In some versions, to avoid long loops, the evolution of $dtm_k = tm_k - tm_{[k-1]}$ is also considered to detect the convergence. The version of the algorithm used in this paper to apply the bisection numerical method is described as follows:

---
**Algorithm A1** Bisection numerical Method
---
1: $t1 \leftarrow \{Ti\}$ Time interval initial value
2: $t2 \leftarrow \{Tf = Ti + Ts\}$ Time interval corresponds to $Ts$, the Time simulation step
3: $V_{ref} \leftarrow \{xc\}$ Reference value to reach and find the time value (solution)
4: $\epsilon \leftarrow \{1E - 15\}$ Tolerance criteria for convergence
5: $error \leftarrow \{1\}$ error as initial value
6: **while** $|error| > \epsilon$ **do**
7:     $tm = (t1 + t2)/2$
8:     $x1 = f(t1)$
9:     $xm = f(tm)$
10:     $error = xm - V_{ref}$
11:     **if** $(x1 - V_{ref}) * (xm - V_{ref}) < 0$ **then**
12:         $t2 = tm$
13:     **else**
14:         $t1 = tm$
15: **end**
16: return $tr = tm$

---

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
