# Peer review of "Nonlinear Dynamics and Performance Analysis of a Buck Converter with Hysteresis Control"

_computation, doi:10.3390/computation9100112_

Round 1
Reviewer 1 Report
Report on paper "Nonlinear Dynamics and Performance Analysis of a Buck Converter with Hysteresis Control" submitted by Velasco et al., for publication in Computation (computation-1378114).
The authors presented the mathematical modeling and experimental implementation of buck converter with a hysteresis control. Although the paper is interesting and contains theoretical and experimental results, it cannot be accepted in its present form and the authors must perform some modifications by addressing the following comments:
1- In the introduction, the literature survey lacks of references in the field of nonlinear dynamics and hysteresis control, which is a topic deeply investigated in the recent past.
2- In subsection 2.5, the implemented continuation method should be described in details and compared to other methods from the literature.
3- The solving procedure should be described in more details including convergence criteria and stability analysis.
4- In subsection 3.2.2, figure 21 is missing and it is not clear how the chaotic behavior can be exploited within the proposed control strategy. Is it possible to exploit such behavior to enhance the device performances?
5- In the conclusion, the limitations of the proposed control strategy should be specified from a critical point of view.
6- The quality of several figures should be enhanced.
Author Response
Reply to Reviewer 1
We want to thank the anonymous reviewer # 1 for his/her revision that led to a further improvement of our manuscript. Replies for all queries presented by reviewer # 1 are given as following:

Reviewer 2 Report
There are many inaccuracies in the manuscript and the power electronic terminology is must be improved!
- Line 14: What does variable structure mean?
- Line 21: This part of the sentence is not clear: “its presence conduct to inaccuracy and instability problems [7].”
- Line 39: PWML is not defined as an abbreviation.
- Line 42: Same for FPIC
- Line 74: “reduction” – It is known as step-down forward converter.
- “According to the status of the switch and diode, three topologies can be obtained as shown in Fig. 2 [24]. In particular, Fig.2(a) shows the first topology in continuous conduction mode (CCM) with positive inductor current iL > 0, which occurs when the switch S is closed, yielding a positive current through the inductor. Hence, the resultant circuit is reduced to the source with the input voltage Vin, an inductor L in series with the source, a shunt capacitor C, and a shunt load resistance R. Additionally, Fig 2(b) shows the second topology evolving in CCM mode with iL > 0, which occurs when the switch S is open. Then, the inductor current continues to flow through the diode and polarizing it to work in direct operation, till the inductor current becomes zero. “ This paragraph is wrong. The authors must familiarize themselves with the CCM, BCM and DCM and rewrite this paragraph. At fixed input and output voltage, these modes are solely dependable on the loading (R)!
- Line 95: “discontinuous system that can be described using a piecewise smooth mathematical” – It is a nonlinear system that is linearized by а piecewise mathematical model!
- Line 103: “polarized” the diode is conducting or reverse biased, but never polarized!
- Line 132: replace “analogical” with a better term!
- Line 134: replace “resistive sensor” It is a resistive divider, not a resistive sensor! Replace it with a better term.
- Line 198: “the power transistor must be de-energized, using” The power electronic switch is turned off! Energising and de-energising is done with capacitors and inductors!
- Line 279: “because the transistorized” The correct term is discrete!
- Table 2: What is the motivation for using these component values? Inductors of in the range of mH (10.6mH) are extremely hard to find, especially for large current. How the values of L and C are calculated (what is the objective?) The input capacitance (GS) of the used transistor is 550pF and the maximum gate current is limited to 1.5mA by two 10k resistors. Considering the fact that the gate voltage is limited to 15V, the rise time of the gate voltage is 5.5us, while the gate signals must be delivered by a driver that provides high peak current to charge and discharge the gate and thus the switching performance of the switch is not deteriorated. In addition to that, P-channel MOSFETS are slower than N-channel MOSFETS. Hence, why a P-channel is used? The authors show very little understanding of the power electronics concept!
- Figure 21 is missing.
- Line 382, “high switching and joule losses” – perhaps if the components selection was made in a better way, the negative impact of the high frequency could be avoided! There are SiC and GaN transistors that could operate at 1MHz (and higher) switching frequency with tolerable switching losses. At these frequencies, the hysteresis control will excel compared to other controls. Please elaborate more on this!
Author Response
Reply to Reviewer 2
We want to thank the anonymous reviewer # 2 for his/her revision that led to a further improvement of our manuscript. Replies for all queries presented by reviewer # 2 are given as following:

Reviewer 3 Report
Please see the comment.
This paper presents a hysteresis control-based output voltage regulator for buck converters, which
suffers from a lack of contributions for both the academic and practical view points; in particular,
1. There are no sufficient reasons to choose the hysteresis control, instead of extant PWMbased
continuous controls
2. There are no new analysis results, such as invariance set, region of attraction, and so on.
Therefore, this paper cannot be published in the current form. The detailed comments are given as
follows.
1. (Section 1) An easy implementation is not sufficient for the proposed hysteresis controller
to be considered as an alternative of the extant PWM-based continuous controllers. This
section has to be comprehensively rewritten with the inclusion of a sufficient comparison
between the proposed and extant techniques (continuous and discontinuous controllers).
2. (Section 2.1) A more bifurcation analysis has to be included under the discontinuous
ON/OFF actions; A Lyapunov analysis may incorporated in this section, leading to the
invariance set and region of attraction-related results. The current analysis is not rigorous,
depending on the numerical simulations.
3. (Section 2.2) This section seems to present the proposed discontinuous controller. However,
there are no sufficient practical merits. There is only one design parameter (switching
frequency). Is this sufficient in practice?
4. (Section 2.6~2.7, Section 3) The simulation and experimental results are not convincing to
show the proposed technique, which are vague due to the absence of the comparison study.
5. (Minor comments)
A. In all paragraph, the mathematical symbol must be distinguished with the text. Please
see the recently published paper in this journal and follow the sample manuscript format.
B. There are numerous typos and formatting issues.
Author Response
Reply to Reviewer 3
We want to thank the anonymous reviewer # 3 for his/her revision that led to a further improvement of our manuscript. Replies for all queries presented by reviewer # 3 are given as following:

Reviewer 4 Report
This paper proposes a mathematical modeling and experimental implementation of a buck converter. The mathematical modeling is well described. However, I have the following questions, which need to be addressed before publication.
1) The authors should write their affiliation in the English language.
2)The authors should spell out the acronym the first time it is used in the body of the manuscript (line 37, 39), What is the PMW signal? (line 118).
3)please check (d) line 60, what is (a) ?.
line 85 is it CCM or DCM? it should be DCM. same with line 100.
line 151 where is Figure 4(b) in the manuscript?
line 336 where is Table 5?
line 341, 342, 346, 352, 354 where is Figure 21, and 21(b)?
4) Please add some references to the previous literature in line 160.
5) Figure 5(b), Figure 19(c),(d), Figure 23, and 24 have no description in the body of the paper, please describe.
Please check figure 19 title,
6) Please add some figures for the experimental board and experimental set-up next to figure 17.
7) Figure 22 (a) is not clear, please make the figures more clear.
8) Please rearrange the 2.1 part to easier follow and understand.
Author Response
Reply to Reviewer 4
We want to thank the anonymous reviewer # 4 for his/her revision that led to a further improvement of our manuscript. Replies for all queries presented by reviewer # 4 are given as following:

Round 2
Reviewer 1 Report
The authors have addressed my comments sufficiently to recommend publication of the paper in its current form.
Author Response
We want to thank the anonymous reviewer # 1 for his/her revision that led to a further improvement of our manuscript. We attach the final paper, additionally more corrections agreed by the authors were made.

Reviewer 2 Report
All comments of the reviewer are addressed accordingly. There are no further questions nor comments to the authors. Hence, the reviewer suggest the manuscript to be accepted.
Author Response
We want to thank the anonymous reviewer # 2 for his/her revision that led to a further improvement of our manuscript. We attach the final paper, additionally more corrections agreed by the authors were made.

Reviewer 3 Report
The revised manuscript may be published in this journal subject to the following minor suggestions:
- please cite the closely related results, such as
- Cascade-Type Pole-Zero Cancellation Output Voltage Regulator for DC/DC Boost Converters, Energies, 2021
- Proportional-Type Sensor Fault Diagnosis Algorithm for DC/DC Boost Converters Based on Disturbance Observer, Energies 2019
- Disturbance Observer-Based Offset-Free Global Tracking Control for Input-Constrained LTI Systems with DC/DC Buck Converter Applications, Energies 2020 - please revised the whole manuscript to check the typos and formatting issue, again.
Author Response
We want to thank the anonymous reviewer # 3 for his/her revision that led to a further improvement of our manuscript. We attach the final paper, additionally more corrections agreed by the authors were made.

Reviewer 4 Report
Thanks for the prompt responses of the authors. In general the manuscript has been improved. one concerns is remaining, in the line 132 the authors should change the PMW to PWM.
Author Response
We want to thank the anonymous reviewer # 4 for his/her revision that led to a further improvement of our manuscript. We attach the final paper, additionally more corrections agreed by the authors were made.
